# Intelligent TCP Congestion Control Policy Optimization

Hanbing Shi  and Juan Wang *

College of Mechatronics Engineering, Xi'an University of Architecture and Technology, Xi'an 710055, China; shbing0601@163.com
* Correspondence: juanwang618@126.com

**Abstract:** Network congestion control is an important means to improve network throughput and reduce data transmission delay. To further optimize the network data transmission capability, this research suggests a proximal policy optimization-based intelligent TCP congestion management method, creates a proxy that can communicate with the real-time network environment, and abstracts the TCP congestion control mechanism into a partially observable Markov decision process. Changes in the real-time state of the network are fed back to the agent, and the agent makes action commands to control the size of the congestion window, which will produce a new network state, and the agent will immediately receive a feedback reward value. To guarantee that the actions taken are optimum, the agent's goal is to obtain the highest feedback reward value. The state space of network characteristics should be designed so that agents can observe enough information to make appropriate decisions. The reward function is designed through a weighted algorithm that enables the agent to balance and optimize throughput and latency. The model parameters of the agent are updated by the proximal policy optimization algorithm, and the truncation function keeps the parameters within a certain range, reducing the possibility of oscillation during gradient descent and ensuring that the training process can converge quickly. Compared to the traditional CUBIC control method, the results show that the TCP-PPO$_2$ policy reduces latency by 11.7–87.5%.

**Keywords:** network congestion; congestion control; internet; proximal policy optimization

## 1. Introduction

With the accelerated growth of mobile broadband network technology and the increase in the number of users in recent years, the huge amount of network information transmission will cause network congestion, and network congestion may lead to a slow transmission speed, high delay, high loss rate, etc., and seriously lead to network failure. To realize the reliable transmission of network data, it is necessary to build an efficient and reliable network transmission protocol, and congestion control is the key technology to achieve efficient and reliable transmission.

NewReno [1] and CUBIC [2] use packet loss to detect congestion and reduce the congestion window length after congestion is detected. Westwood [3] is an adaptation of NewReno, and the transmission capacity-based congestion control mechanism uses the prediction of the link transmit capacity as the basis for congestion control.

Traditional congestion control mechanisms use defined control rules to adjust congestion windows, making it difficult to adapt to the complexity and real-time changes in modern networks. Therefore, the researchers propose a congestion control algorithm based on reinforcement learning. Van et al. used reinforcement learning algorithms to adaptively change parameter configurations [4], thereby improving the quality of the video stream. Cui et al. proposed a custom congestion control algorithm Hd-TCP [5], which applies deep reinforcement learning to deal with the poor network experience caused by frequent network switching on a high-speed rail from the perspective of the transport layer. Lin et al. improved the applicability of virtual network functions using a model-assisted deep reinforcement learning framework [6]. Xie et al. proposed a congestion window length

for 5G mobile edge computing based on deep learning [7]. TCP-Drinc is a model-free intelligent congestion control algorithm based on deep reinforcement learning [8], which obtains eigenvalues from past network states and experiences and adjusts the congestion window length based on the set of these eigenvalues. The Rax algorithm uses online reinforcement learning [9] to maintain the optimal congestion window length based on the given reward function and network conditions. QTCP is based on Q-learning for congestion control [10,11], which improves throughput to a certain extent. MPTCP [12] uses Q-learning and Deep Q-Networks (DQN) for multipath congestion control, which is able to learn to take the best action based on the runtime state. However, the Q-learning algorithm is slow to learn and difficult to converge. The reinforcement learning algorithm based on the policy gradient can solve the shortcomings of the Q-learning algorithm such as slow learning speed and difficult convergence.

To further improve the communication capability of the congestion control strategy in the unknown network environment, this paper analyzes the characteristics of the four stages of congestion control and proposes a congestion control strategy based on the proximal policy optimization algorithm, which is one of the best policy gradient algorithms [13]; this strategy can save a lot of model training time, make full use of the training data, and finally realize the reliable transmission of data. Compared with the traditional CUBIC congestion control strategy, the proposed algorithm is feasible and effective in improving network transmission performance.

## 2. Related Work

### 2.1. Fundamentals of Congestion Control

Network congestion is a phenomenon that often occurs in the operation of computer networks, and from its manifestation, network congestion is the phenomenon that the cache in the router drops packets because of overflow. When the packet arrives at the router, the packet is forwarded according to the configured forwarding rules and output to the corresponding link. Due to limited network link resources (including cache size, fixed bandwidth, processing power, etc.), queues form in the link and network congestion occurs when packets arrive too quickly. Increasing the buffer area can absorb excess packets and prevent packet loss, but if you blindly increase the size of the buffer without improving the link bandwidth and processor capacity, this will cause the waiting time in the queue to greatly increase, and the upper protocol can only retransmit them, so simply expanding the cache space can not solve the network congestion problem, but will cause a waste of network resources. In addition, network nodes process thousands of data streams per second, sharing bandwidth between data streams, and the maximum rate of data transmission is limited by the bottleneck link. Congestion occurs when a network node needs to process more data than it can handle. Therefore, the job of congestion control is to prevent data senders from sending large amounts of data into the network, causing the transmission link to be overloaded. The principle of congestion control is shown in Figure 1.

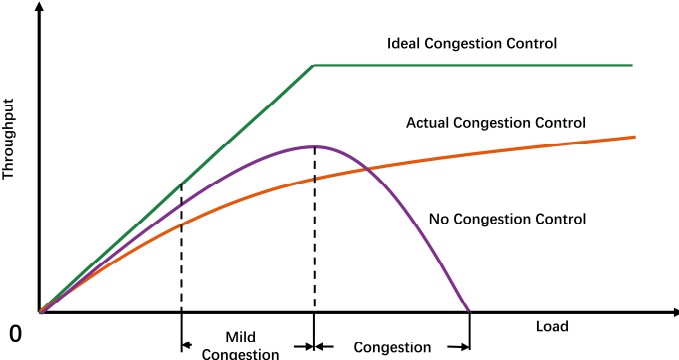

**Figure 1.** The role of congestion control.

### 2.2. Deep Reinforcement Learning-Based Congestion Control

The congestion control strategy framework based on deep reinforcement learning is shown in Figure 2. Deep reinforcement learning requires the construction of environments and agents. Taking the network environment as the environment, by collecting the real-time state of the network environment, the strategy function used by the agent is constructed, the agent responds after learning, and the strategy function is fitted by an artificial neural network. The agent makes the optimal control strategy according to the output of the policy function, controls the congestion window length, and changes the TCP sending policy. After the agent makes an action, deep reinforcement learning will judge the action according to the state, so as to output the reward value and depending on the reward value, modify the parameters of the artificial neural network so that the agent can maximize the reward.

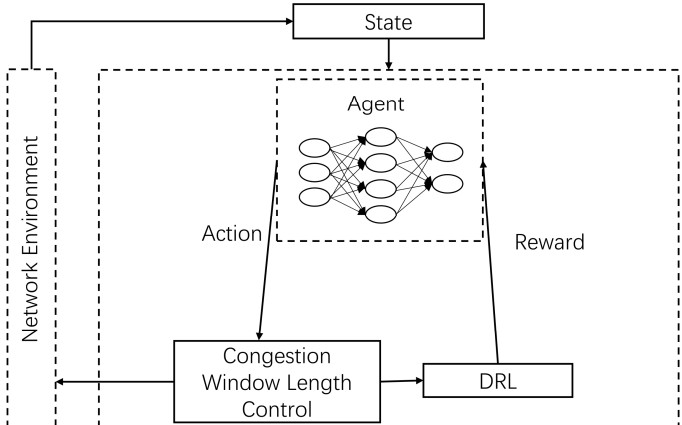

**Figure 2.** DRL-based congestion control algorithm framework.

The QTCP algorithm is an algorithm based on the Q-learning framework. Its state space is continuous, and its state includes average RTT, average interval time between sending and receiving packets, and discrete action spaces including increasing by 10 bytes, decreasing by 1 byte, and remaining unchanged. Next, the system compares the size of the utility function of the current time period and the previous time period to determine whether it is a positive reward or a negative reward, where the utility function $U = a \cdot \log(throughput) - b \cdot \log(RTT)$. QTCP algorithms perform better than traditional algorithms in a variety of network scenarios, but the effect is still linked to network scenarios and cannot completely avoid the shortcomings of traditional congestion control algorithms. The congestion control algorithm Indigo, based on imitation learning, sets the network scene information to expert knowledge, and the decision network is a single-layer LSTM network, to achieve congestion control in the current training scenario. However, the performance of the algorithm can only play a superior performance in the trained scenario, so the practical application is limited. In contrast, DRL-based congestion control algorithms only require a simple neural network and combine historical information based on multiple time slice states before the current moment to obtain performance beyond traditional algorithms, so we will take a closer look at DRL-based congestion control algorithms.

## 3. Methods

### 3.1. Deep Reinforcement Learning

#### 3.1.1. Background

Deep reinforcement learning is a combination of deep learning and reinforcement learning. Deep learning uses representation learning to refine data [14] and does not have to choose features, compressed dimensions, conversion formats, or other data processing techniques, offering better feature representation capabilities than conventional machine learning techniques and providing a distributed representation of data by mixing low-level

features to create more abstract high-level features. Reinforcement learning originated from the optimal control theory in cybernetics [15], which is mainly used to solve the problem of timing decision making, by ongoing environmental interaction and trial-and-error, and finally obtains the optimal strategy for a specific task and maximizes the cumulative expected return of the task. The mainstream methods of traditional reinforcement learning mainly include the Monte Carlo class method and the time difference classification method [16]. The former is an unbiased estimate with a larger variance, while the latter uses a finite step bootstrapping method with a smaller variance but introduces bias. Deep reinforcement learning combines the structure of deep learning with the idea of reinforcement learning for solving decision-making problems. With the help of the powerful representation ability of deep neural networks, any component of reinforcement learning can be fitted, including state value functions, action value functions, strategies, models, etc., and the weights in deep neural networks can be used as fitting parameters. DRL is mainly used to solve high-dimensional state action space tasks, integrating deep learning's powerful understanding ability in feature representation problems and reinforcement learning's decision-making ability to achieve end-to-end learning. The emergence of deep reinforcement learning has made reinforcement learning technology truly practical and it can solve complex problems in real-world scenarios. The most representative deep Q-network (DQN) is an extension of the Q-learning algorithm [17], which uses neural networks to approximate the action–value function, and the optimization goal is the minimization loss function:

$$L_i(\theta_i) = E_{\pi_{\theta_i}}[(y_i - Q(s, a; \theta_i))^2] \tag{1}$$

where $i$ represents the ith iteration, $y_i = r + \gamma \max_{a'} Q(s', a'; \theta_{i-1})$.

Van et al. proposed the use of the deep double Q-network (DDQN) [18]. DDQN selects the action of the target Q value based on the current Q-network and uses the target Q-network to calculate the corresponding Q value of the action.

$$y_i = r + \gamma Q(s', \text{argmax}_a Q(s', a; \theta_i); \theta_i^-) \tag{2}$$

### 3.1.2. Proximal Policy Optimization Algorithms

The description of reinforcement learning is usually based on the Markov decision process [19], which is a mathematical formalization of sequential decision making, in which immediate rewards and subsequent states of the system are affected by behavior, causing changes in future rewards, whose tasks correspond to the multivariate array E = <S, A, P, R>, where:

S—is a state space.
A—is an action space.
P—is state transition probability.
R—is the reward function.

Cumulative discount returns are often used to define state returns at t moment:

$$R_t = \sum_{i=t}^{T} \gamma^{i-1} r(s_i, a_i) \tag{3}$$

$\gamma$ is the discount factor, which indicates that the farther away the return, the less impact it has on the assessment of the current state, $r(s_i, a_i)$ represents the value of the return obtained by selecting the action $a_i$ in the state $s_i$; the initial state is $s_i$, and under a certain policy $\pi$, the state distribution obeys $\rho_\pi$, then the task of reinforcement learning is to learn a policy $\pi$ so that the desired initial state return is maximized.

### 3.1.3. Policy Gradient

The Policy Gradient (PG) method works by calculating the estimator of the strategy gradient and inserting it into the stochastic gradient ascent algorithm. The most commonly used gradient estimators have this form:

$$\hat{g} = \hat{E}_t[\nabla_\theta \log \pi_\theta(a_t|s_t)\hat{A}_t] \tag{4}$$

where $\pi_\theta$ is the stochastic strategy and $\hat{A}_t$ is the estimator of the dominant function at time step $t$. Here, the expectation $\hat{E}_t[\ldots]$ represents the empirical mean of the finite batch sample, in algorithms alternating between sampling and optimization, using the implementation of automatic discrimination software to work by constructing an objective function with a gradient as a gradient estimator of the strategy gradient. By deriving the target, the estimator $\hat{g}$ is obtained.

$$L^{PG}(\theta) = \hat{E}_t[\log \pi_\theta(a_t|s_t)\hat{A}_t] \tag{5}$$

While it is advantageous to use a uniform trajectory to perform multiple optimization steps for $L^{PG}$, it is not reasonable and, empirically, often leads to a large number of policy updates being disrupted.

### 3.1.4. Trust Region Methods

In TRPO, the objective function is maximized but is limited by the size of the policy update. The details are as follows:

$$\underset{\theta}{\max imize}\,\hat{E}_t\Big[\frac{\pi_\theta(a_t|s_t)}{\pi_{\theta_{old}}(a_t|s_t)}\hat{A}t\Big] \tag{6}$$

$$Subject\ to\ \hat{E}_t[KL[\pi_{\theta_{old}}(\cdot|s_t), \pi_\theta(\cdot|s_t)]] \le \delta \tag{7}$$

$\theta_{old}$ is the vector of policy parameters before the update. After linear approximation of the target and quadratic approximation of the constraint, the conjugate gradient algorithm can be used to effectively approximate the problem.

This theory proves that TRPO actually recommends using penalties rather than constraints, i.e., solving unconstrained optimization problems.

$$\underset{\theta}{\max imize}\,\hat{E}_t\left[\frac{\pi_\theta(a_t|s_t)}{\pi_{\theta_{old}}(a_t|s_t)}\hat{A}t - \beta KL[\pi_{\theta_{old}}(\cdot|s_t), \pi_\theta(\cdot|s_t)]\right] \tag{8}$$

The coefficients $\beta$ follow the fact that a defined agent targets from the lower bound of the strategy $\pi$, and TRPO uses hard constraints rather than penalties because it is difficult to choose a single $\beta$ value that performs well in different problems or single problems, where features change during the learning process. Therefore, it is not enough to simply choose a fixed penalty coefficient $\beta$ using SGD to optimize the penalty target (8).

### 3.1.5. Proximal Policy Optimization Algorithm

The Proximal Policy Optimization (PPO) algorithm, which is one of the most effective model-free policy gradient methods, achieves state-of-the-art performance in many reinforcement learning continuous control benchmarks [13]. It is derived from the TRPO algorithm [20], but is easier to implement and has better sample complexity. Therefore, PPO is used to train a defined congestion-controlled reinforcement learning strategy. PPO is an actor-critical algorithm, so it uses a multi-step return of TD($\lambda$) as a function of training values and a generalized advantage estimator (GAE) to compute the policy gradient [21].

### 3.1.6. PPO$_2$ Principle

The reinforcement learning algorithm needs to design the strategy function $\pi(a_t|s_t)$ so that it can generate the probability of performing some action $a_t$ under state $s_t$. Artificial neural networks can theoretically fit arbitrary functions, so the current reinforcement

learning algorithm fits the strategy function $\pi(a_t|s_t)$ through artificial neural networks, and the neural network parameters are denoted as $\theta$. The goal of reinforcement learning is to make each action achieve the maximum reward value, and the core is how to judge the quality of the selected action. To do this, the following advantage function is defined:

$$\hat{L}_{\pi_\theta,t} = \hat{R}_t - V_\phi(s_t) \tag{9}$$

where $V_\phi(s_t)$ is a function of the value of the state and reflects all the cumulative reward values that are expected to be achieved after the end of the round under state $s_t$. The dominance function reflects the advantage of selecting an action $a_t$ relative to the average action $a_t$ at moment t. If the value $v_t$ corresponding to all states s and action a is a two-dimensional table, the large value range of states $s_t$ will cause the storage space of the two-dimensional table to be large and difficult to store. Therefore, the artificial neural network is also chosen to approximate the value function $V_\phi(s_t)$. Finally, the optimization objective function for reinforcement learning is defined as follows:

$$L^{MSE} = E_{\pi_{\theta_k}}(\hat{R}_t - V_\phi(s_t))^2 \tag{10}$$

The goal of the (10) function is to update the strategy function parameter $\theta$ so that each action can obtain a larger reward value. However, the problem with the objective function $L^{MSE}$ is that if the parameter $\theta$ is updated too much, it will cause repeated oscillations when the gradient rises without fast convergence to the best advantage. For this reason, the PPO$_2$ algorithm redefines the following objective function formula:

$$L^{clip}(\theta) = \hat{E}_t[\min(r_t(\theta)\hat{A}_t, clip(r_t(\theta), 1-\varepsilon, 1+\varepsilon)\hat{A}_t)] \tag{11}$$

The clip function is a truncation function, defined as:

$$clip(r, 1-\varepsilon, 1+\varepsilon) = \begin{cases} r & 1-\varepsilon < r < 1+\varepsilon \\ 1-\varepsilon, & r \leq 1-\varepsilon \\ 1+\varepsilon, & r \geq 1+\varepsilon \end{cases} \tag{12}$$

$r_t(\theta)$ is a probability ratio function, defined as:

$$r_t(\theta) = \frac{\pi_\theta(a_t|s_t)}{\pi_{\theta_{old}}(a_t|s_t)} \tag{13}$$

$r_t(\theta)$ reflects the magnitude of parameter updates, the larger $r_t(\theta)$ is, the larger the amplitude of the update parameter, and vice versa. The goal of Equation (10) is to obtain a biased estimate of the value function $V_\phi(s_t)$, so the commonly used least squares method is used to define the objective function, and the squared operation ensures that the objective function is non-negative. When the value of the advantage function in Equation (11) is positive, it means that the reward value obtained by the current action is higher than the average, and the objective function optimization goal is to let the agent choose such actions as much as possible. When the dominance function is negative, it means that the reward value obtained by the current action is lower than the average, and the agent should avoid selecting this action. The $L^{clip}(\theta)$ function avoids excessive update fluctuations by intercepting $r_t(\theta)$ to limit it to $[1-\varepsilon, 1+\varepsilon]$. The $L^{clip}(\theta)$ function is schematically shown in Figure 3. When the dominant function $L > 0$ (Figure 3a), if $r_t(\theta)$ is greater than $1+\varepsilon$, it is truncated so that it is not too large. Similarly, when $L < 0$ (Figure 3b), if $r_t(\theta)$ is less than $1-\varepsilon$, it is also truncated so that it is not too small. The $L^{clip}(\theta)$ function (Figure 3c) guarantees that $r_t(\theta)$ does not fluctuate sharply.

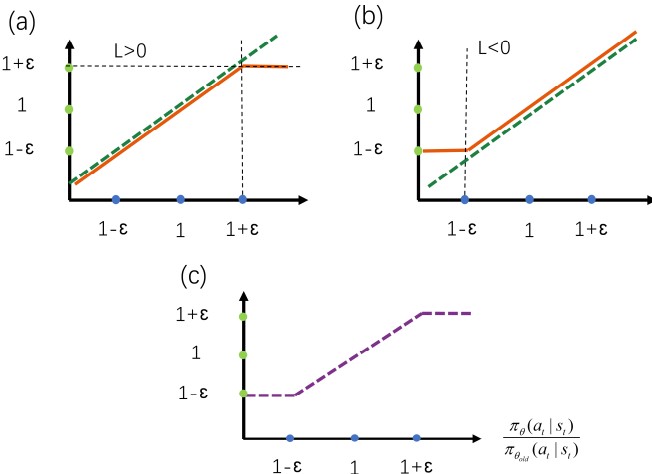

**Figure 3.** Schematic diagram of the truncation function; (**a**) is the range of values when the value of the dominant function > 0; (**b**) is the range of values when the value of the dominant function < 0; and (**c**) is the final range of values allowed by the truncation function.

### 3.2. State Space Design

A reasonable state $s_t$ is crucial for efficient reinforcement learning implementation, and only by observing enough information can the reinforcement learning algorithm make the correct action choice. Excessive state information can lead to slower learning and increased computational demands. Therefore, this paper refers to the state parameters and design state $s_t$ required by mainstream TCP algorithms such as CUBIC for decision making. $s_t$ contains the following parameters.

(1) The current relative time $t_r$. Described as the amount of time that has passed when TCP first established the connection up to the present. In algorithms such as CUBIC, the window length is designed as a third-degree function of time $t_r$. Consequently, $t_r$ plays a crucial role in determining the congestion window.

(2) The size of the current congestion window. The adjustment of the new window value in the congestion control algorithm should be based on the present congestion window length, which can be increased at a faster rate if the current congestion window length is small, and stopped or increased more slowly if the window is large.

(3) The number of bytes is not acknowledged. Defined as the number of bytes transmitted but not yet acknowledged by the receiver. The unacknowledged bytes can be metaphorically compared to water stored in a pipe, where the network link is similar to the pipe. This parameter is also an important parameter that the congestion control algorithm needs to refer to. If the amount of water in the pipe is sufficient, it should stop or reduce the injection of water into the pipe; if the amount of water in the pipe is small, the amount of water injection into the pipe should be increased, and the volume of water in the pipe may be used to calculate the water injection rate (congestion window duration).

(4) The quantity of ACK packets obtained. When the normal amount of ACK packets is received, the network is functioning well without congestion and the congestion window length can gradually be increased. If the network is congested, with a reduced number of ACK packets received, the congestion window length should either be kept constant or reduced.

(5) RTT. Latency refers to the total time it takes for a packet to be sent to the receiving acknowledgment packet, which can be figuratively understood as the time it takes for the data to make a round trip from the sender to the receiver. Network congestion and latency are strongly associated, and when network congestion is bad, latency increases a lot. As a result, the delay can be an indicator of network congestion, and

the congestion control algorithm can modify the congestion window in response to the delay.

(6) Throughput rate. Described as the number of data bytes the receiver acknowledges each second. A high throughput rate indicates that enough packets have been transmitted in the present connection; alternatively, it shows that there is more available network capacity and that more packets may be sent to the link. This parameter directly reflects the network circumstances.

(7) The number of packet losses. The higher the number of packet losses, the more serious the current network congestion is, and the congestion window size needs to be reduced; a small number of packet losses suggests that the current network is not congested, and the congestion window length should be increased.

### 3.3. Action Space Design

$a_t$ is the control action made at the moment t for the congestion window. This document defines the action to increase the congestion window length c by n segment length.

$$c = c_{old} + ns'$$ (14)

The idea of Equation (14) is to provide a generalization formula that determines the rate of growth of the congestion window length based on the observed state parameter information. Different policies should be selected in different network scenarios. In a high-bandwidth environment, $n > 1$ should be adjusted to increase the congestion window length at an exponential rate; in a low-bandwidth environment, $n = 1$ should be adjusted to make the congestion window grow at a linear speed. When network congestion occurs, $n \leq 0$ should be adjusted to maintain or reduce the length of the congestion window and reduce the pressure of network congestion.

### 3.4. Reward Function Design

The reward from the environment at time t is referred to as reward $r_t$, and the design reward letter is as follows:

$$r_t = \alpha \left( \frac{O}{O_{\max}} \right) - (1 - \alpha) \frac{l_{\min}}{l}$$ (15)

where O is the currently observed throughput rate and $O_{\max}$ is the maximum throughput rate observed in history, and the ratio of the two reflects the throughput rate effect that can be increased by the action $a_t$; l represents the average delay during the observation period and $l_{\min}$ represents the smallest delay observed in history, and the ratio of the two reflects the delay effect of action $a_t$ improvement; and $\alpha$, a hyperparameter that measures the weight ratio of throughput rate and delays to the reward, is a weight factor. $\alpha$ defines whether the congestion control algorithm's optimization objective is more concerned with throughput rate or delay. In this example, $\alpha = 0.5$ is selected to balance throughput and latency. In addition, the minimum throughput rate and the maximum delay of the history are saved. When it is observed that the current throughput rate is less than or equal to the minimum throughput rate or greater than or equal to the maximum delay, the reward is set to $-10$ to avoid reaching these two extreme states.

### 3.5. Algorithm Description

The input of the Algorithm 1 is the current state of the network $s_t$, and the output is the congestion window length $c_{new}$. The pseudocode is as follows.

---

**Algorithm 1.** PPO2

---

1. Input: $s_t$ = {congestion window length, the number of ACK packets, latency, throughput rate, packet loss rate}.

2. Initialize the policy parameters $\theta_0 = \theta_{old} = \theta_{new}$.

3. Run strategy $\pi_{\theta k}$ for a total of T time steps, collect $\{s_t, a_t\}$.

4. $\theta_{old} \leftarrow \theta_{new}$

5. $r_t = \alpha \left( \frac{O}{O_{\max}} \right) - (1 - \alpha) \frac{l_{\min}}{l}$.

6. $\hat{R} = \sum\limits_{t=0}^{T} \gamma^t r_t$.

7. $L^{clip}(\theta) = \hat{E}_t[\min(r_t(\theta)\hat{A}_t, clip(r_t(\theta), 1 - \varepsilon, 1 + \varepsilon)\hat{A}_t)]$.

8. Update the parameter θ by the gradient ascent method so that $L^{clip}(\theta)$ is the maximum.

9. Output: The length of the new congestion window after adjustment is $c = c_{old} + ns'$.

---

## 4. Experiment

### 4.1. Experimental Environment

#### 4.1.1. Hardware and Software Environment

The test was performed on a powerful server, and the specific configuration is as follows: ① CPU, AMD Ryzen 7 5800H with Radeon Graphics @3.19 GHz; ② Memory, 16 GB DDR4; ③ GPU, NVIDIA GeForce RTX 3060 Laptop 6 GB; and ④ Operating System, Windows 11.

The data space topology is simulated by the Mininet simulator, and the TCP-PPO$_2$ and TCP-CUBIC algorithms are implemented, which are compared with the traditional TCP congestion control algorithms.

#### 4.1.2. PPO Algorithm Parameter Settings

The main parameters of PPO$_2$ are set as follows: the neural network's hidden layers total 2, the number of neurons in the two layers is 32 and 16, respectively, the discount factor is 0.99, the learning rate is 0.00025, the $\varepsilon$ is 0.2, and the number of training steps when running each update is 128.

### 4.2. Bandwidth Sensitivity Comparison

The amount of data that may be transmitted per unit of time is referred to as network bandwidth (generally 1 s). The greater the bandwidth, the greater its traffic capacity. Figure 4 shows the bandwidth sensitivity comparison, and the link bandwidth in our Mininet is set to between 1 Mbps and 1000 Mbps, with 0 ms latency, 1000 packet queues, and 0% random loss. TCP-PPO$_2$ works well in this bandwidth range; when the link bandwidth is less than 18 Mbps, both traditional and deep reinforcement learning-based congestion control protocols produce large delays, mainly due to the narrow bandwidth and too long queue length causing queuing problems, but when the bandwidth is higher than 50 Mbps, TCP-PPO$_2$ can achieve extremely low latency.

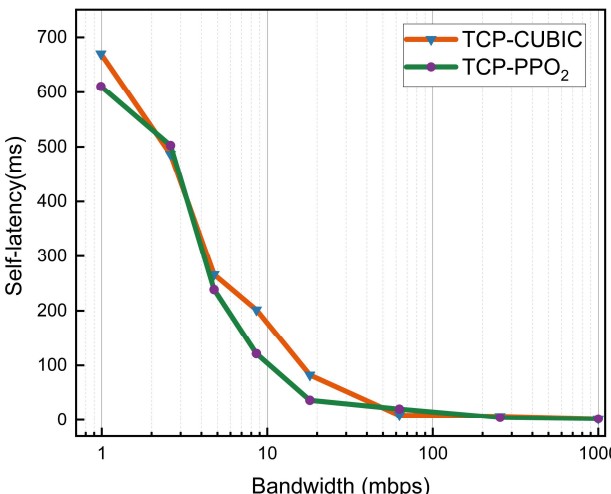

**Figure 4.** Under the condition that only the bandwidth size is changed, comparison of the latency of the traditional congestion control strategy and the congestion control strategy based on deep reinforcement learning.

### 4.3. Latency Sensitivity

RTT represents the time it takes for a packet to be sent from sent to receive to acknowledged, reflecting the current network latency. Figure 5 shows a comparison of latency sensitivity, with the link bandwidth set to 700 Mbps, a queue of 1000 packets, and 0% random loss in our Mininet. When the link delay is less than 50 ms, the system latency of the traditional congestion control protocol is much higher than that of the intelligent congestion control strategy, indicating that TCP-PPO$_2$ can adapt to today's low-latency networks, thereby ensuring more efficient data transmission. When the link delay is higher than 90 ms, the delay of both is greatly reduced, but the delay of the intelligent system is still lower than the system delay of the TCP-CUBIC protocol.

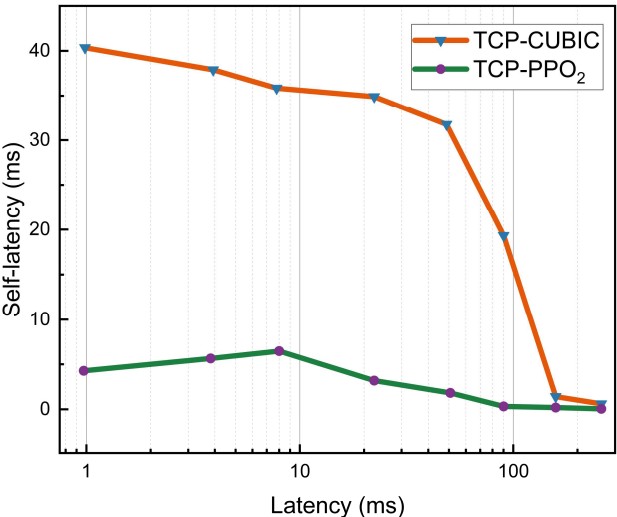

**Figure 5.** Under the condition that only the network latency is changed, comparison of the latency of the traditional congestion control strategy and the congestion control strategy based on deep reinforcement learning.

### 4.4. Queue Sensitivity

The queue represents the size of the packets sent at one time; Figure 6 shows the queue sensitivity comparison, and we changed the queue size between 1 and 10,000 packets. Other configurations of the links in Mininet are a bandwidth of 700 Mbps, a latency of

40 ms, and a random loss of 0%. When the queue size is greater than 10, the delay of the traditional congestion control strategy is significantly higher than that of TCP-PPO$_2$.

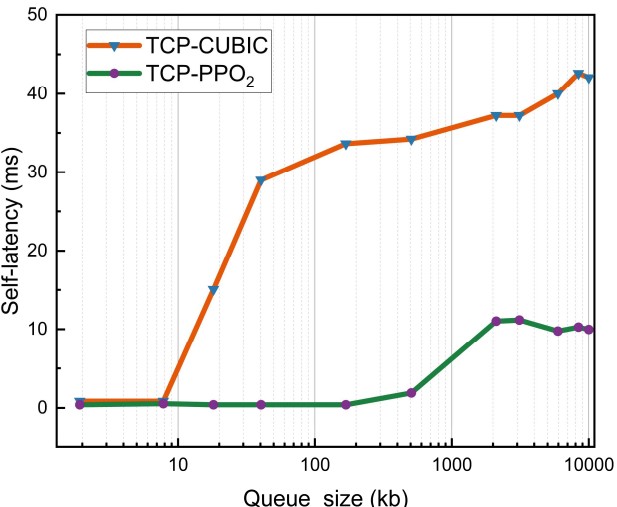

**Figure 6.** Under the condition that only the queue size is changed, comparison of the latency of the traditional congestion control strategy and the congestion control strategy based on deep reinforcement learning.

### 4.5. Packet Loss Sensitivity

The packet loss rate is an important indicator to measure the reliability of network transmission protocols, and Figure 7 shows the packet loss sensitivity comparison, setting a random loss rate of up to 8%. Other configurations of links in Mininet are a bandwidth of 700 Mbps, a latency of 40 ms, and queues of 1000 packets. When the random loss rate increases from zero, the latency of both congestion control strategies decreases rapidly, while the delay at the beginning of TCP-PPO2 is significantly lower than that of traditional congestion control strategies.

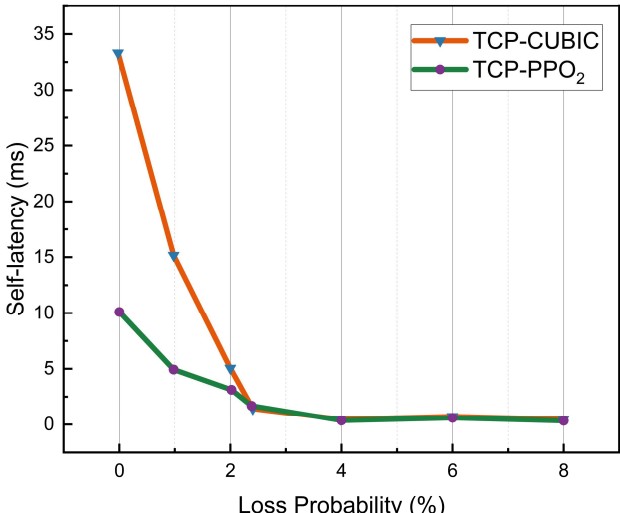

**Figure 7.** Under the condition that only the packet loss rate is changed, comparison of the latency of the traditional congestion control strategy and the congestion control strategy based on deep reinforcement learning.

### 5. Conclusions

Aiming at the problems of poor adaptability of mainstream TCP congestion control algorithms and inability to effectively use virtual data space network borrowing, a TCP

congestion control strategy based on deep reinforcement learning is proposed, which effectively improves the data transmission efficiency. The main conclusions of this paper are as follows:

(1) Optimize the traditional TCP congestion control strategy by using the near-end policy optimization algorithm, map the system's send rate to the behavior of deep reinforcement learning, set the reward function by balancing throughput, latency, and packet loss, and use a simple deep neural network to approximate the final strategy. Through the comparison of a large number of experimental data, the parameters such as the number of neural network layers, the number of neurons, and the length of the history were determined, and the optimization of TCP-PPO2 was successfully realized.

(2) Through Mininet simulation experiments, it is determined that the TCP congestion control algorithm based on the proximity policy optimization adapts to network changes faster than the traditional TCP congestion control algorithm, changes the real-time congestion window size, improves transmission efficiency, and reduces the data transmission delay by 11.7–87.5%.

**Author Contributions:** Conceptualization, H.S.; methodology, H.S.; software, H.S.; validation, H.S. and J.W.; formal analysis, H.S. and J.W.; investigation, H.S. and J.W.; resources, H.S. and J.W.; data curation, H.S.; writing—original draft preparation, H.S.; writing—review and editing, J.W.; visualization, H.S.; supervision, J.W.; project administration, J.W.; funding acquisition, J.W. All authors have read and agreed to the published version of the manuscript.

**Funding:** This research was funded by Xi'an Key Laboratory of Clean Energy: 2019219914SYS014C G036; Key R&D Program of Shaanxi Province: 2023-ZDLGY-24.

**Institutional Review Board Statement:** Not applicable.

**Data Availability Statement:** The data are not publicly available due to privacy.

**Conflicts of Interest:** The authors declare no conflict of interest.

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
