# Peer review of "Intelligent TCP Congestion Control Policy Optimization"

_applsci, doi:10.3390/app13116644_

Round 1
Reviewer 1 Report
Reviewer’s Comments:
The manuscript “Intelligent TCP congestion control policy optimization” is a very interesting work. This paper investigates the network congestion control is an important means to improve network throughput and reduce data transmission delay. To further optimize the network data transmission capability, This research suggests a proximal policy optimization-based intelligent TCP congestion management method. creates a proxy that can communicate with the real-time network environment and abstracts the TCP congestion control mechanism into a partially observable Markov decision process. Changes in the real-time state of the network are fed back to the agent, and the agent makes action commands to control the size of the congestion window, which will produce a new network state, and the agent will immediately receive a feedback reward value. To guarantee that the actions taken are optimum, the agent's goal is to get the highest feedback reward value. Design the state space of network characteristics so that agents can observe enough information to make appropriate decisions. The reward function is designed through a weighted algorithm that enables the agent to balance and optimize throughput and latency. While I believe this topic is of great interest to our readers, I think it needs major revision before it is ready for publication. So, I recommend this manuscript for publication with major revisions.
1. In this manuscript, the authors did not explain the importance of network congestion in the introduction part. The authors should explain the importance of network congestion.
2) Title: The title of the manuscript is not impressive. It should be modified or rewritten it.
3) Correct the following statement “The model parameters of the agent are updated by the proximal policy optimization algorithm, and the truncation function keeps the parameters within a certain range, reducing the possibility of oscillation during gradient descent and ensuring that the training process can converge quickly. Compared to the traditional CUBIC control method, the results show that TCP-PPO2 reduces the delay by 11.7%-87.5% compared to the traditional congestion control strategy”.
4) Keywords: The network congestion is missing in the keywords. So, modify the keywords.
5) Introduction part is not impressive. The references cited are very old. So, Improve it with some latest literature.
6) The authors should explain the following statement with recent references, The agent makes the optimal control strategy according to the output of the policy function, controls the congestion window length, and changes the TCP sending policy”.
7) Add space between magnitude and unit. For example, in synthesis “21.96g” should be 21.96 g. Make the corrections throughout the manuscript regarding values and units.
8) The author should provide reason for this statement “The unacknowledged bytes can be metaphorically compared to water stored in a pipe, where the network link is similar to the pipe”.
9. Comparison of the present results with other similar findings in the literature should be discussed in more detail. This is necessary in order to place this work together with other work in the field and to give more credibility to the present results.
10) Conclusion part is very long. Make it brief and improve by adding the results of your studies.
11) There are many grammatic mistakes. Improve the English grammar of the manuscript.
Minor editing of English language required
Author Response
Dear reviewers and editors,
We are very grateful to the editors and all reviewers for taking the time to provide positive and constructive comments. These opinions are valuable and helpful for the revision and improvement of our "Research on Congestion Control Strategy Based on Deep Reinforcement Learning" (ID: applsci-2371163), and have important guiding significance for our research. We revised the manuscript based on the reviewers' comments.
For more information about modifications, see Attachments.
We hope that our response will be satisfactory and that the manuscript will be accepted for publication.
Thank you so much for taking the time on this manuscript!
Sincerest regards
Professor Wang Juan

Reviewer 2 Report
- Add a notation table to explain the meaning of all the notation used in the paper because various terms used in the Q-learning need to be clarified.
- Check all the headings of the paper and check the spelling as well.
- Nothing is added under Heading 3.1.
- Various notations need to be appropriately defined, and the authors must invest some time to add details on the missing terms.
N/A
Author Response

(The authors gave the same response as above.)

Reviewer 3 Report
The introduction mixes several problems and does not provide a clear statement what is done in the paper. First it tells about the need for the client to detect the congestion control algorithm used by the server, then it tells about developing a new congestion control, which is a completely different problem. Thus most of the introduction is not related with the rest of the paper.
The referenced papers are too old, the latest one is from 2017.
The novelty of this research is my major concern, because there are numerous papers about TCP + Q-learning and even TCP + deep Q-learning. The proposed solution should be compared with them, otherwise it is unclear how is it better than the existing solutions.
Author Response

(The authors gave the same response as above.)

Round 2
Reviewer 3 Report
My main concern regarding the contribution of the paper in comparison with the existing research is not resolved. The authors added some references to the published research that solves a similar problem and designs a reinforcement learning-based algorithm to adjust congestion window. However, these solutions are not compared with the one proposed in the paper. Thus I cannot judge if the proposed solution is better than them.
For example, in this paper we see that the proposed solution improves latency, but in the solutions in these papers:
Xiao, Kefan, Shiwen Mao, and Jitendra K. Tugnait. "TCP-Drinc: Smart congestion control based on deep reinforcement learning." IEEE Access 7 (2019): 11892-11904.
Jay, Nathan, et al. "A deep reinforcement learning perspective on internet congestion control." International Conference on Machine Learning. PMLR, 2019.
also improve latency, so which one is better?
Author Response

(The authors gave the same response as above.)
